# Comparative Assessment of the Addictive Potential of Synthetic Cathinones by Zebrafish Conditioned Place Preference (CPP) Paradigm

**DOI:** 10.3390/life14070820

**Published:** 2024-06-27

**Authors:** Liao-Chen Chen, Ming-Huan Chan, Hwei-Hsien Chen

**Affiliations:** 1Center for Neuropsychiatric Research, National Health Research Institutes, 35 Keyan Road, Miaoli County 35053, Taiwan; angeleks.tan@gmail.com; 2Institute of Systems Neuroscience, College of life Science and Medicine, National Tsing Hua University, 101, Section 2, Kuang Fu Road, Hsinchu 30044, Taiwan; 3Institute of Neuroscience, National Chengchi University, 64, Section 2, ZhiNan Road, Wenshan District, Taipei City 11605, Taiwan; 4Department of Medical Research, China Medical University Hospital, 2, Yude Road, North District, Taichung City 404327, Taiwan

**Keywords:** reward, extinction, reinstatement, pentylone, eutylone, N-ethylpentylone

## Abstract

Synthetic cathinones have gained increasing popularity in the illicit drug market, yet their abuse potential remains poorly understood. In this study, zebrafish were used to compare the addictive potential of three cathinone analogs, namely pentylone, eutylone, and N-ethylpentylone (NEP). The zebrafish received various doses (0 to 60 mg/kg) of the cathinone analogs by oral gavage over two sessions per day for two consecutive days to induce conditioned place preference (CPP). Pentylone, eutylone, and NEP dose-dependently induced CPP, with NEP showing significantly higher CPP than pentylone and eutylone at the dose of 20 mg/kg. The fish that received 60 mg/kg of cathinones underwent extinction, followed by reinstatement triggered by drug priming. NEP required six sessions to meet the criteria of extinction, followed by eutylone, which required four sessions, and pentylone, which required three sessions. Furthermore, NEP and eutylone at a dose of 40 mg/kg could reinstate the extinguished CPP, while 60 mg/kg of pentylone was necessary for CPP reinstatement. The persistence of susceptibility to reinstatement was also assessed at 7 and 14 days after the initial reinstatement. The CPP induced by all three cathinone analogs could be reinstated 7 days after the initial reinstatement, whereas only CPP induced by NEP, but not pentylone and eutylone, could be reinstated again after 14 days. Considering the potency to induce CPP, resistance to extinction, and the propensity for reinstatement, the abuse liability rank order of the cathinone analogs might be as follows: NEP > eutylone > pentylone. These findings suggest that the zebrafish CPP paradigm can serve as a viable model for assessing the relative abuse liability of substances.

## 1. Introduction

Cathinone, also known as β-keto amphetamine is extracted from khat (Catha edulis), a perennial shrub native to the Arabian Peninsula, Eastern African countries, and China [1]. Synthetic cathinones, which are cathinone derivatives with various structural modifications designed to avoid legal restrictions, are often categorized into different generations based on their development and chemical structure modifications over time. β-keto-methylenedioxyamphetamines (i.e., novel psychoactive substances with names ending in “ylone”) are commonly found in “Ecstasy” formulations due to the presence of a methylenedioxy functional group on the phenyl ring, similar to MDMA. Pentylone belongs to the second generation of synthetic cathinones. Eutylone and N-ethylpentylone (NEP), which are derived from pentylone and are part of the third generation of cathinones, have recently emerged in illicit drug markets and recreational settings worldwide.

Pentylone and eutylone are structural isomers. Pentylone features a propyl group at the α-carbon and a methyl group at the amine position. Eutylone, on the other hand, has ethyl groups at the α-carbon and amine positions. N-ethylpentylone, as its name suggests, possesses a propyl group at the α-carbon and an ethyl group at the amine position (Figure 1).

Pharmacologically, pentylone, eutylone, and NEP are efficacious inhibitors of monoamine transporters. Pentylone and eutylone exhibit nearly identical potency in inhibiting the dopamine transporter (DAT) and norepinephrine transporter (NET). Eutylone is slightly more potent than pentylone as a serotonin transporter (SERT) inhibitor [2]. The potencies of NEP in inhibiting monoamine transporters are much higher than those of pentylone and eutylone. However, pentylone and eutylone act as SERT substrates with a partial releasing effect, while NEP is devoid of substrate and transporter releasing activities [3].

The adverse symptoms associated with these three cathinones are similar and include psychomotor agitation, aggressiveness, confusion, tachycardia, visual hallucinations, hyperthermia, psychosis, inconsistent speech, sleeplessness, and cardiac arrest [4]. Fatal cases of intoxication from accidental overdose have been reported [5,6].

Behavioral studies in animals demonstrate that pentylone, eutylone, and NEP are effective locomotor stimulants [2,7,8,9,10]. Additionally, acute administration of NEP induces anxiolytic effects, aggressive behavior, and social exploration deficits in mice. NEP also induces depressive-like symptoms after repeated administrations [11,12].

The rewarding effects of pentylone and NEP have been studied using the conditioned place preference (CPP) paradigm in mice [13] and eutylone in rats [14]. However, determining their relative addictive potential from these assessments is challenging. The CPP paradigm, where animals associate the drug (unconditioned stimulus, US) with their environment (conditioned stimulus, CS), has been utilized to investigate the rewarding effects of psychoactive compounds. In addition, following the acquisition and subsequent extinction of CPP, animals often reinitiate this response, a phenomenon known as reinstatement, which reflects craving, a core feature of drug addiction [15]. Extinction also serves as a useful additional measure to quantify the rewarding effect of drugs in rats [16]. Resistance to extinction reflects an increasing function of rewarding. However, the considerable amount of additional time and effort required for measuring the extinction and reinstatement of CPP in rodents makes it less suitable for comparing the addictive potential of different drugs.

Compared to rats or mice, zebrafish (Danio rerio) husbandry and behavioral research facilities are relatively simple and economical. Zebrafish share a significant genetic homology and conservation of neurotransmitter and neuroendocrine systems with humans [17]. CPP has been adopted in zebrafish to study drug reward. Zebrafish have demonstrated CPP responses to a variety of drugs of abuse, including nicotine [18], ethanol [19], opioid compounds [20], and psychostimulants [21]. Recently, a comprehensive methamphetamine (MA) CPP model, including acquisition, expression, extinction, and reinstatement, which showed similarities to rodent models was successfully established in adult zebrafish [22]. This suggests that the zebrafish CPP model is well-suited for evaluating the addictive potentials of synthetic cathinones.

The present study was designed to evaluate the relative addictive potential of pentylone, eutylone, and NEP using the CPP model, including acquisition, expression, extinction, and reinstatement, in zebrafish. Firstly, we examined the dose-dependent effects of these three cathinones on CPP. Secondly, we assessed the number of sessions required to reach the criteria of extinction at the same effective dose to determine their resistance to extinction. Finally, we compared their priming dose needed to induce CPP reinstatement and re-induced CPP reinstatement after forced abstinence for 7 or 14 days to determine their propensity to reinstatement. Based on these measurements, the rank order of the addictive potential of these three cathinone analogs might be as follows: NEP > eutylone > pentylone.

## 2. Materials and Methods

### 2.1. Animals and Housing

This study utilized zebrafish of the AB wild strain aged between 3 and 6 months. The zebrafish were obtained from the Taiwan Zebrafish Core Facility at the National Health Research Institutes. The fish were housed in a controlled environment at a temperature of 28 °C, with a light cycle of 14 h of light and 10 h of darkness. They were fed zebrafish powder food twice a day (Zeigler, Pentair Aquatic Eco-Systems, Inc., Cary, NC, USA). All experimental procedures involving the zebrafish were conducted in accordance with the approved guidelines and were approved by the Institutional Animal Care and Use Committee of the National Health Research Institutes, Taiwan (NHRI-IACUC-109049-M1-A-S01).

### 2.2. Drugs

Pentylone and N-ethylpentylone (NEP) (95%) were provided by the Taiwan Taoyuan District Prosecutors Office. Eutylone (95%) was provided by the Taiwan Kaohsiung District Prosecutors Office. Ethyl 3-aminobenzoate methanesulfonate salt (Tricaine) was purchased from Sigma-Aldrich (St. Louis, MO, USA). All drugs were dissolved in distilled water.

### 2.3. CPP Paradigm

The CPP paradigm was as previously described [22]. Prior to the experiment, the zebrafish were weighed and those fish weighing approximately 0.5 g were selected and individually housed, to ensure accurate identification, in plastic tanks measuring 16 cm (L) × 11 cm (W) × 10 cm (H) for two consecutive days. The CPP apparatus consisted of a tank measuring 44 cm (L) × 8 cm (W) × 14 cm (H) with a divider separating two compartments as follows: one white and one navy with yellow vertical stripes. During the pre-conditioning testing phase, each zebrafish was placed in the CPP apparatus without a divider and allowed to acclimate for the initial 10 min. The percentage of time spent by the fish in each compartment of the CPP tank was then measured over the subsequent 10 min.

After the pretest session, zebrafish underwent training sessions twice daily for two consecutive days. An oral gavage procedure was utilized to administer precise quantities of drugs. Each zebrafish was placed in the preferred compartment (white) for 20 min after receiving water, and in the non-preferred compartment (navy with yellow vertical stripes) after receiving a dose of pentylone, eutylone, or NEP. On the day following conditioning, each zebrafish was placed in the CPP apparatus without a divider and allowed to explore for 15 min. After a 5 min acclimatization period, the percentage of time spent by the fish on each side of the CPP apparatus was measured for 10 min.

After the CPP test, non-confined extinction was performed. During the extinction session, zebrafish received a daily 15 min CPP test. This continued until the fish reached the extinction criteria. After the completion of the extinction training, the fish were administered a priming dose of either pentylone, eutylone, or NEP via oral gavage. Subsequently, they were placed in the CPP apparatus to reinstate the previously extinguished CPP. Similar to the testing phase, after a 5 min acclimatization period, the percentage of time spent by the fish on each side of the CPP apparatus was measured for 10 min.

In the case of assessment of the persistent susceptibility to reinstatement, a retest and drug-primed reinstatement of CPP were sequentially conducted after a period of 5 days of abstinence following the initial reinstatement. This procedure was repeated after another 5 days of abstinence.

### 2.4. Statistical Analyses

Data were expressed as mean ± S.E.M. The dose-dependent effects of pentylone, eutylone, and NEP were analyzed by two-way ANOVA, with dose and drug as factors. When the same animals were repeatedly tested at different phases, one-way repeated measures ANOVA was used. A post hoc Tukey’s comparison test was used. *p* < 0.05 was considered statistically significant.

## 3. Results

### 3.1. Dose-Dependent Effects of Pentylone, Eutylone, and NEP CPP

During the conditioning phase of CPP, zebrafish received various doses (20, 40, and 60 mg/kg) of pentylon, eutylone, or NEP to assess their dose-dependent effects. As shown in Figure 2A, the percentage of time spent on the drug-paired side during pre-test was similar in all groups. The data for the CPP test (Figure 2B) were analyzed using a two-way ANOVA, revealing significant drug and dose effects, as well as a significant drug × dose interaction (drug: F _2,60_ = 7.72, *p* < 0.01; dose: F _3,60_ = 65.36, *p* < 0.001; interaction: F _6,60_ = 3.68, *p* < 0.01). All three cathinones induced CPP at 40 and 60 mg/kg. However, eutylone and NEP, but not pentylone, exhibited CPP at 20 mg/kg. There was no significant difference between the three cathinones at the dose of 60 mg/kg. Thus, this dose was selected to induce CPP for the subsequent experiments.

### 3.2. Extinction of Pentylone, Eutylone, and NEP CPP

After CPP was induced by pentylone, eutylone or NEP at 60 mg/kg, zebrafish underwent daily non-confined extinction sessions until they met the extinction criteria. The extinction criterion was considered met when the percentage of fish spending time on the drug-paired side equaled the pre-test level. As depicted in Figure 3, Figure 4 and Figure 5, the fish conditioned with the different cathinones required varying numbers of extinction sessions to successfully achieve the extinction criterion. NEP required six sessions to meet the criteria, followed by eutylone, which required four sessions, and pentylone, which required three sessions. The number of extinction sessions required to meet the criteria is associated with the resistance to extinction.

### 3.3. Drug-Primed Reinstatement of Pentylone, Eutylone, and NEP CPP

Upon reaching the extinction criterion, the fish were initially administered the vehicle first to ensure no CPP reinstatement via oral gavage. The next day, 40 mg/kg was used as a drug priming dose to reinstate the CPP for the three cathinones. NEP and eutylone at 40 mg/kg significantly reinstated the extinguished CPP. Pentylone (40 mg/kg) failed to induce CPP reinstatement. Although eutylone (40 mg/kg) could induce reinstatement of CPP, this dose did not reach the level observed during the CPP test. Subsequently, pentylone and eutylone at 60 mg/kg were applied, resulting in a higher reinstatement of CPP response. The effective priming dose of pentylone was higher than that of eutylone and NEP.

### 3.4. Effects of Long Abstinence on NEP, Pentylone, and Eutylone CPP Expression and Reinstatement

After a period of 5 days of abstinence following the initial reinstatement, the retests and drug-primed reinstatement of CPP were sequentially conducted. As shown in Figure 3, Figure 4 and Figure 5, during the retest session, all the zebrafish spent a similar or equivalent amount of time on the drug-paired side compared to the pre-test phase. Pentylone (60 mg/kg), eutylone (60 mg/kg), and NEP (40 mg/kg) could significantly induce reinstatement of CPP.

Subsequently, the retests and drug-primed reinstatement of CPP were conducted again after another 5 days of abstinence. At this time, there was still no CPP response during retesting. However, only the NEP-induced CPP response could be reinstated, whereas pentylone and eutylone failed to reinstate CPP response. It appeared that NEP-induced CPP showed a higher susceptibility to reinstatement compared to pentylone and eutylone.

## 4. Discussion

The present study evaluated the relative abuse liability of pentylone, eutylone, and NEP through the measurement of the dose-dependent effects of CPP induction, resistance to extinction, and propensity to reinstatement in zebrafish. Our findings revealed that pentylone, eutylone, and NEP were all capable of inducing CPP in zebrafish, with the effectiveness varying based on the dosage administered. At a concentration of 60 mg/kg, pentylone, eutylone, and NEP elicited nearly identical effects on CPP outcomes. However, at lower dose (20 mg/kg) NEP-induced the strongest CPP, followed by eutylone. Pentylone (20 mg/kg) failed to induce CPP. These results suggest that NEP may have a higher potency than pentylone and eutylone in producing rewarding effects.

The resistance to extinction was compared when CPP was induced by pentylone, eutylone, and NEP at the same dose (60 mg/kg). Our results showed that NEP-induced CPP in zebrafish required six extinction training sessions, while eutylone- and pentylone-induced CPP necessitated four and three sessions, respectively, to meet the extinction criteria under non-confined extinction training. Resistance to extinction of the lever-pressing response is dependent on both the magnitude of the reward (large or small) and the schedule of reinforcement (continuous or partial) [23]. Similarly, resistance to extinction of CPP response is related to the strength of conditioning, which may, in turn, be related to the rewarding efficacy of a drug [16]. Under the same CPP procedure and training dose, the rank order of resistance to extinction was NEP > eutylone > pentylone, potentially reflecting their rewarding efficacy.

The reinstatement of CPP, which resembles drug craving and subsequent drug seeking behavior, is associated with memory retrieval. Extinguished CPP can be reinstated by a priming dose of the drug. The susceptibility to CPP reinstatement is associated with persistent drug-associated memory, potentially linked to the drug’s rewarding efficacy. NEP and eutylone at a dose of 40 mg/kg could reinstate the extinguished CPP, while 60 mg/kg of pentylone was required to induce CPP reinstatement. Additionally, we examined the sustained susceptibility to CPP reinstatement induced by these three cathinone analogs. CPP induced by all three cathinone analogs could be reinstated 7 days after the initial reinstatement. However, only NEP-induced CPP could be reinstated again 14 days after the initial reinstatement. It appears that a higher priming dose of pentylone was needed to reinstate CPP, and NEP-induced CPP exhibited a longer sustained susceptibility to reinstatement. Therefore, the rank order of susceptibility to reinstatement is NEP > eutylone > pentylone, providing additional insight into the measurement of drug rewarding effects.

It has been reported that among drugs with mixed actions on monoamine systems, those which potently interact with DAT are associated with high abuse liability [24], whereas those drugs which display higher potency at SERT compared to DAT are associated with low abuse potential [25,26]. Moreover, the abuse potential of mixed-action synthetic cathinones has been shown to increase with repeated exposure, possibly due to the sustained DAT-mediated effect while tolerance develops to the SERT-mediated effects that initially limit the abuse potential of these compounds [27]. Accordingly, DAT/SERT selectivity has been used as an index of abuse liability. NEP functions solely as a pure uptake inhibitor, exhibiting the highest DAT/SERT selectivity among the three cathinones, which is consistent with our findings of its greatest resistance to extinction and susceptibility to reinstatement.

In addition to the inhibition of transporters, pentylone and eutylone also act as SERT substrate releasers. A comparison of the pharmacological properties of pentylone and eutylone revealed that eutylone exhibits equal potency for DAT inhibition and higher potency for SERT uptake inhibition. However, it displays lower potency in inducing serotonin release via SERT compared to pentylone in rat brain synaptosomes [2]. In addition to the uptake inhibition and release stimulation, recent advancements in transporter pharmacology have uncovered emerging complexities, such as partial releasers [28], allosteric inhibitors [29], and atypical transporter ligands [30]. In fact, pentylone and eutylone have displayed weak partial releasing actions at SERT [2]. While the DAT/SERT ratio has been used to predict abuse liability, applying it to compare the relative abuse potential of pentylone and eutylone poses challenges.

The bioavailability of the cathinones to the CNS might also contribute to their potency in the CPP response. It is well recognized that cathinones can freely cross the blood–brain barrier (BBB), with NEP showing higher permeability than pentylone in rats [7]. Similar to mammals, zebrafish develop a functional BBB. The higher BBB permeability of NEP may contribute to its increased potency in CPP response and its greater abuse liability.

A comparative evaluation of the rewarding effects of pentylone and NEP in a mouse CPP paradigm showed that both substances induced a significant increase in preference score at 3 and 10 mg/kg, whereas NEP, but not pentylone, elicited CPP at 30 mg/kg [13]. In zebrafish, NEP induced a significantly greater preference than pentylone at 20 mg/kg, with consistent results at 40 and 60 mg/kg. While the profiles of CPP induced by pentylone and NEP in mice and zebrafish differed, both animal models demonstrated NEP may produce rewarding effect over a broad dose range.

The present study successfully developed a zebrafish CPP model to assess the potency for CPP induction, susceptibility to extinction, and propensity for reinstatement as an initial screen for comparison of the relative abuse liability of cathinones. Complementary studies in mammals such as an assessment of the breakpoint by self-administration under progressive ratio schedules of reinforcement [26] or using dose-response analysis under a simple fixed ratio schedule and behavioral economic procedures [31] are still needed to ensure a more comprehensive understanding of their relative abuse liability.

Indeed, the same zebrafish CPP paradigm, including the extinction and reinstatement phases, was applied for MA in our previous study [22]. It took seven extinction sessions to fulfill the extinction criteria in MA-induced CPP at a dose of 40 mg/kg. At a dose of 20 mg/kg, MA could reinstate CPP. Moreover, susceptibility to reinstatement persisted after abstinence for 14 days. Based on its higher resistance to extinction and longer sustained susceptibility to reinstatement than NEP, it is suggested that the abuse liability of MA surpasses that of NEP. These findings align with the relative reinforcing effects of MA and NEP observed in self-administration and behavioral economics analyses in rats [32].

Zebrafish maintenance and experimentation are relatively inexpensive compared to studies involving mammals, making them a cost-effective model for evaluating the abuse liability of drugs. However, several species differences between zebrafish and mammals may limit the direct applicability of the findings to humans. Firstly, the complexity and organization of the zebrafish brain differ significantly from those of mammals. These neurological differences could impact how drugs affect processes related to addiction. Additionally, zebrafish are agastric and the pH of the intestines never reaches below 7.5 under homeostatic conditions [33], which could influence drug absorption and biotransformation when administered orally.

The pharmacodynamic and pharmacokinetic profiles of drugs can vary greatly between species, making it challenging to determine the appropriate comparative doses. Therefore, the doses used in the zebrafish may not directly translate to equivalent doses in humans or other animals. Future experiments should include pharmacokinetic studies to better understand the absorption, distribution, metabolism, and excretion (ADME) profiles of these drugs in zebrafish.

## 5. Conclusions

In summary, the present study successfully assessed the relative abuse liability of pentylone, eutylone, and NEP based on the CPP-inducing potency, resistance to extinction, and propensity for reinstatement in zebrafish. The results suggest that the rank order of abuse liability is NEP > eutylone > pentylone. Additionally, these findings suggest that zebrafish CPP induction, extinction, and reinstatement can serve as a viable model for assessing the abuse liability of substances, at least for cathinones and psychostimulants. This highlights the utility of zebrafish as a valuable tool in preclinical drug abuse research.

## Figures and Tables

**Figure 1 life-14-00820-f001:**
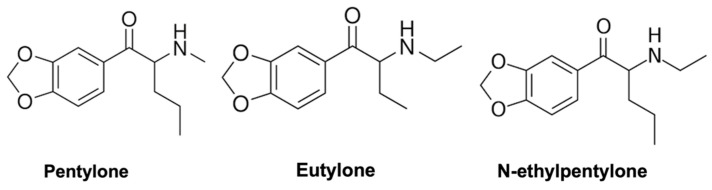
Chemical structures of pentylone, eutylone, and NEP.

**Figure 2 life-14-00820-f002:**
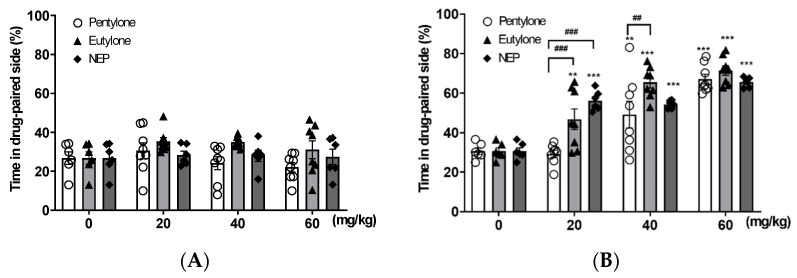
Dose-response effects of pentylone, eutylone, and NEP on CPP. Different doses of pentylone (0, 20, 40, or 60 mg/kg) were administered before CPP conditioning. The percentage of time spent in the drug-paired side was recorded during both the pretest (**A**) and the CPP test (**B**). The data for CPP test were analyzed using a two-way ANOVA with drug and dose as factors (dose: F_3,60_ = 65.36, *p* < 0.001; drug: F_2,60_ = 7.72, *p* < 0.01; interaction: F _6,60_ = 3.68, *p* < 0.01). Data were expressed as mean ± SEM (*n* = 6–8). ** *p* < 0.01, *** *p* < 0.001, compared with respective control (0 mg/kg), ^##^ *p* < 0.01, ^###^
*p* < 0.001 compared with the group indicated.

**Figure 3 life-14-00820-f003:**
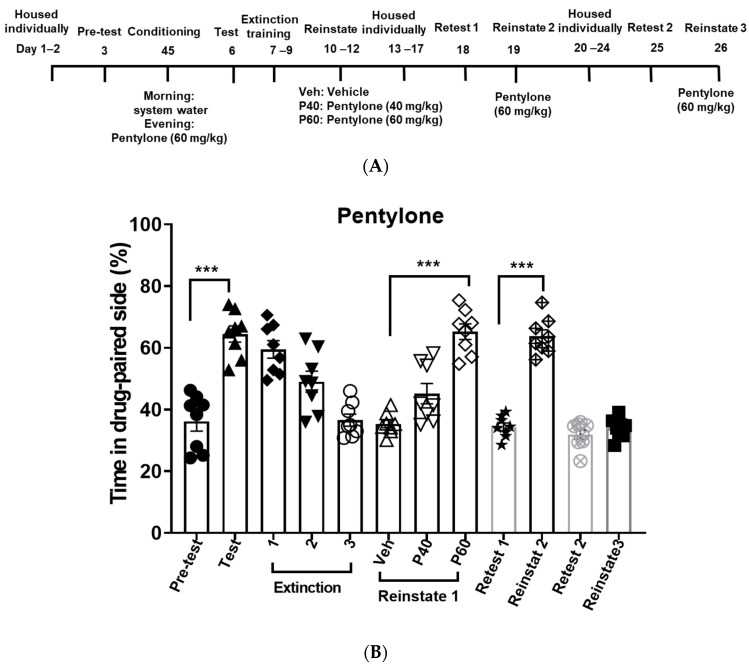
Resistance to extinction and propensity for reinstatement of CPP induced by pentylone. Experimental timeline (**A**) and percentage of the time in drug-paired side for each session (**B**). Zebrafish were conditioned with pentylone (60 mg/kg) followed by non-confined extinction sessions until reaching the criteria. Then, the effective priming doses to induce reinstatement were determined by applying doses of 40 mg/kg or 60 mg/kg. Following the initial reinstatement, a retest and drug-primed reinstatement of CPP were sequentially conducted after a period of 5 days of abstinence. The same procedure was repeated after another 5 days of abstinence. The number of extinction sessions required to meet the criteria indicates the susceptibility to extinction. The effective priming dose and persistent vulnerability to induce reinstatement reflect the propensity for reinstatement. Data were expressed as mean ± SEM (*n* = 8). The data were analyzed using one-way repeated measures ANOVA, with post hoc Tukey’s test. *** *p* < 0.001, compared with respective control.

**Figure 4 life-14-00820-f004:**
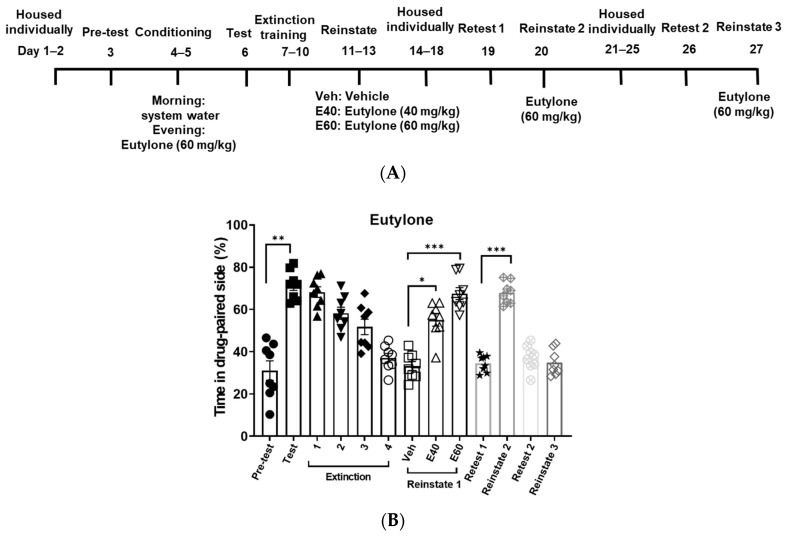
Resistance to extinction and propensity for reinstatement of CPP induced by eutylone. Experimental timeline (**A**) and percentage of the time in drug-paired side for each session (**B**). Eutylone 60 mg/kg was used to induce CPP. The experimental protocol was the same as described in Figure 3. Data were expressed as mean ± SEM (*n* = 8). The data were analyzed using one-way repeated measures ANOVA, with post hoc Tukey’s test. * *p* < 0.05, ** *p* < 0.01, *** *p* < 0.001, compared with the session indicated.

**Figure 5 life-14-00820-f005:**
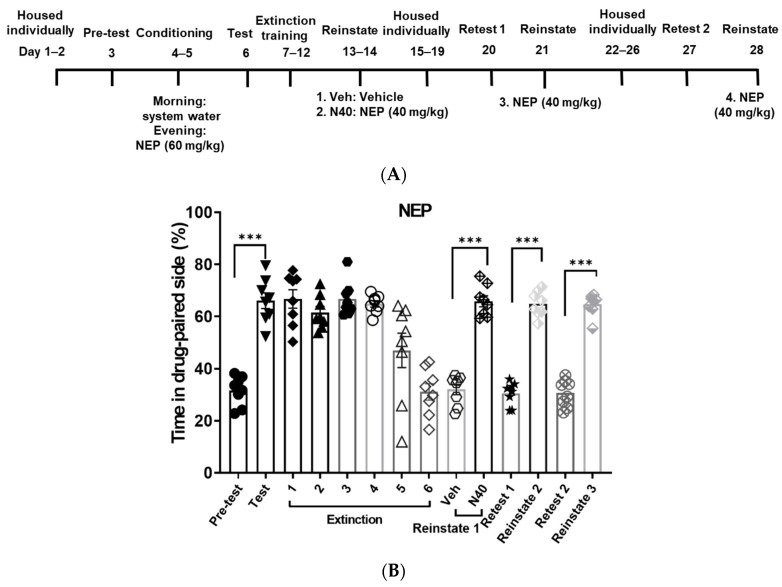
Resistance to extinction and propensity for reinstatement of CPP induced by NEP. Experimental timeline (**A**) and percentage of the time in drug-paired side for each session (**B**). NEP (60 mg/kg) was used to induce CPP. The experimental protocol was the same as described in Figure 3. Data were expressed as mean ± SEM (*n* = 8). The data were analyzed using one-way repeated measures ANOVA, with post hoc Tukey’s test. *** *p* < 0.001, compared with the session indicated.

## Data Availability

The data that support the findings of this study are available on request from the corresponding author.

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
