# Peer review of "Comparative Assessment of the Addictive Potential of Synthetic Cathinones by Zebrafish Conditioned Place Preference (CPP) Paradigm"

_life, 2024, doi:10.3390/life14070820_

Round 1

Reviewer 1 Report

Comments and Suggestions for Authors

1. on what basis did the authors choose the exact dose of 20, 40 and 60 mg of each drug in order to perform a dose-response relationship?

2. as zebrafish are agastric and has pH neutral, while the drugs were given orally, is it possible that the results obtained are associated with poor/high absorption of the drugs tested or differences in biotransformation?

3. Introduction should be shortened, however the authors should focus on potential differences between the drugs tested.

4. Please provide the limitations of the study

5. didi the authors determine the bioavailability of the drugs to the central nervous system? This may be helpful to determine whether a lack of CPP ar altered CPP is due to a lack of or altered bioavailability of the drug  

Comments on the Quality of English Language

minor changes are required 

Author Response

Thank you very much for taking the time to review this manuscript. Please find the detailed responses below and the corresponding revisions highlighted in the re-submitted files.

Comment 1: on what basis did the authors choose the exact dose of 20, 40 and 60 mg of each drug in order to perform a dose-response relationship?

Response 1: In the conditioned place preference (CPP) study of rodents, three cathinone analogs were able to induce CPP at a dose of 30 mg/kg. Based on this, the initial dose of 20 mg/kg was chosen and was extended to 40 and 60 mg/kg to assess dose-dependent effects.

Comment 2: as zebrafish are agastric and has pH neutral, while the drugs were given orally, is it possible that the results obtained are associated with poor/high absorption of the drugs tested or differences in biotransformation?

Response 2: We acknowledge that zebrafish are agastric and the pH of the intestines never reaches below 7.5 under homeostatic conditions, which could influence drug absorption and biotransformation when administered orally. The results obtained could indeed be affected by these factors. To address this, future experiments should include pharmacokinetic studies to better understand the absorption, distribution, metabolism, and excretion (ADME) profiles of these drugs in zebrafish. This issue has been added in the discussion on page 9.

Comment 3: Introduction should be shortened, however the authors should focus on potential differences between the drugs tested.

Response 3: As suggested, we have revised and shorten the Introduction section to enhance focus and readability. The revised introduction will concentrate more specifically on the potential differences between the drugs tested.

Comment 4: Please provide the limitations of the study

Response 4: As suggested, the limitations of current study have been added on page 9 as follows.

However, several species differences between zebrafish and mammals may limit the direct applicability of the findings to humans. Firstly, the complexity and organization of the zebrafish brain differ significantly from those of mammals. These neurological differences could impact how drugs affect processes related to addiction. Additionally, zebrafish are agastric and the pH of the intestines never reaches below 7.5 under homeostatic conditions [33], which could influence drug absorption and biotransformation when administered orally.

The pharmacodynamic and pharmacokinetic profiles of drugs can vary greatly be-tween species, making it challenging to determine appropriate comparative doses. Therefore, the doses used in zebrafish may not directly translate to equivalent doses in humans or other animals. Future experiments should include pharmacokinetic studies to better understand the absorption, distribution, metabolism, and excretion (ADME) profiles of these drugs in zebrafish. 

Comment 5: did the authors determine the bioavailability of the drugs to the central nervous system? This may be helpful to determine whether a lack of CPP or altered CPP is due to a lack of or altered bioavailability of the drug.

Response 5: The bioavailability of the drugs to the central nervous system (CNS) was not specifically determined in this study. However, we have added a paragraph in the discussion section on page 9 to address this issue.

The bioavailability of the cathinones to the CNS might also contribute to their potency in CPP response. It is well recognized that cathinones can freely cross the blood–brain-barrier (BBB), with NEP showing higher permeability than pentylone in rats [7]. Similar to mammals, zebrafish develop a functional BBB. The higher BBB permeability of NEP may contribute to its increased potency in CPP response and its greater abuse liability.

Reviewer 2 Report

Comments and Suggestions for Authors

The study presented in this article offers a thorough and compelling evaluation of the relative abuse liability of three cathinone analogs—pentylone, eutylone, and N-ethylpentylone (NEP)—using a zebrafish conditioned place preference (CPP) model. This research contributes significantly to our understanding of the addictive potential of these substances and highlights the utility of zebrafish as a viable and cost-effective model for assessing drug abuse liability.

One of the notable strengths of this study is its detailed examination of dose-dependent effects on CPP induction, resistance to extinction, and propensity for reinstatement. The findings that NEP induces the strongest CPP at lower doses compared to pentylone and eutylone, and requires more sessions for extinction, provide important insights into the higher potency and rewarding efficacy of NEP. Additionally, the sustained susceptibility to reinstatement observed with NEP further underscores its potential for higher abuse liability.The comparative approach taken in this study, particularly the use of the same doses across different cathinones, allows for a clear and direct comparison of their effects. This methodical design strengthens the conclusions drawn regarding the rank order of abuse potential, which is convincingly supported by the data: NEP > eutylone > pentylone. The discussion also appropriately contextualizes these findings within the broader framework of drug abuse research, citing relevant studies and pharmacological properties that align with the observed outcomes.

In conclusion, this study makes a valuable contribution to the field of drug abuse research. The use of zebrafish CPP paradigms offers a promising alternative for preliminary screening of abuse potential, and the insights gained from this research pave the way for more extensive investigations using diverse models. Future research should focus on validating these findings across different species and exploring the long-term behavioral and neurological impacts of these substances. Overall, this article is a significant step forward in the ongoing effort to understand and mitigate the risks associated with synthetic cathinone abuse.

While the study presented in the article provides valuable insights into the abuse liability of pentylone, eutylone, and N-ethylpentylone (NEP) using zebrafish, there are several potential issues and limitations that should be considered:

  1. Species Differences: The extrapolation of findings from zebrafish to humans or even other mammals can be problematic. While zebrafish offer a cost-effective and convenient model, their neurobiology and pharmacokinetics differ significantly from those of mammals, potentially limiting the applicability of the results to human drug abuse scenarios.
  2. Dose Translation: The doses used in zebrafish may not directly translate to equivalent doses in humans or other animals. The pharmacodynamic and pharmacokinetic profiles of drugs can vary greatly between species, making it challenging to determine appropriate comparative doses.
  3. Behavioral Complexity: Zebrafish, despite being a useful model organism, have simpler behavioral repertoires compared to mammals. This simplicity may not capture the full complexity of drug-induced behaviors observed in higher-order animals, which can impact the interpretation of CPP, extinction, and reinstatement data.
  4. Long-term Effects: The study primarily focuses on the short-term effects of the cathinone analogs. Long-term studies are needed to assess the chronic impacts and potential neurotoxicity of these substances. Longitudinal studies in mammalian models would provide a more comprehensive understanding of the long-term risks associated with these drugs.
  5. Pharmacological Differences: The study mentions the differences in pharmacological actions, such as DAT/SERT selectivity and SERT substrate releasing properties. However, it does not delve deeply into how these differences might affect other aspects of drug abuse liability, such as tolerance, sensitization, or cross-sensitization with other drugs.

Author Response

Thank you for the reviewer’s insightful comments. The potential issues and limitations you've highlighted regarding our study on the abuse liability of pentylone, eutylone, and N-ethylpentylone (NEP) using zebrafish have been thoroughly considered. Here is our response to each point:

Comment 1: Species Differences: The extrapolation of findings from zebrafish to humans or even other mammals can be problematic. While zebrafish offer a cost-effective and convenient model, their neurobiology and pharmacokinetics differ significantly from those of mammals, potentially limiting the applicability of the results to human drug abuse scenarios.

Response 1: We agree that species differences pose a significant challenge when extrapolating findings from zebrafish to humans or other mammals. We have addressed this issue on page 9 as follows.

However, several species differences between zebrafish and mammals may limit the direct applicability of the findings to humans. Firstly, the complexity and organization of the zebrafish brain differ significantly from those of mammals. These neurological differences could impact how drugs affect processes related to addiction. Additionally, zebrafish are agastric and the pH of the intestines never reaches below 7.5 under homeostatic conditions [33], which could influence drug absorption and biotransformation when administered orally.

Comment 2: Dose Translation: The doses used in zebrafish may not directly translate to equivalent doses in humans or other animals. The pharmacodynamic and pharmacokinetic profiles of drugs can vary greatly between species, making it challenging to determine appropriate comparative doses.

Response 2: Translating doses from zebrafish to humans or other animals is indeed complex due to differences in pharmacokinetics and pharmacodynamics across species. As suggested, a new paragraph has been added in the discussion as follows.

The pharmacodynamic and pharmacokinetic profiles of drugs can vary greatly between species, making it challenging to determine appropriate comparative doses. Therefore, the doses used in zebrafish may not directly translate to equivalent doses in humans or other animals. Future experiments should include pharmacokinetic studies to better understand the absorption, distribution, metabolism, and excretion (ADME) profiles of these drugs in zebrafish.

Comment 3: Behavioral Complexity: Zebrafish, despite being a useful model organism, have simpler behavioral repertoires compared to mammals. This simplicity may not capture the full complexity of drug-induced behaviors observed in higher-order animals, which can impact the interpretation of CPP, extinction, and reinstatement data.

Response 3: Zebrafish behavior is undoubtedly less complex than that of mammals, which may limit the scope of behavioral analyses. While our study utilized the zebrafish conditioned place preference paradigm as an initial screen for abuse potential, we recognize the need for complementary studies in mammals. The related description has been added on page 9 as follows.

The present study successfully developed a zebrafish CPP model to assess the potency for CPP induction, susceptibility to extinction, and propensity for reinstatement, as an initial screen for comparison of the relative abuse liability of cathinones. Complementary studies in mammals such as assessment of the breakpoint by self-administration under progressive ratio schedules of reinforcement [26] or using dose-response analysis under a simple fixed ratio schedule and behavioral economic procedures [31], are still needed to ensure a more comprehensive understanding of their relative abuse liability.

Comment 4: Long-term Effects: The study primarily focuses on the short-term effects of the cathinone analogs. Long-term studies are needed to assess the chronic impacts and potential neurotoxicity of these substances. Longitudinal studies in mammalian models would provide a more comprehensive understanding of the long-term risks associated with these drugs.

Response 4: The present study was intended as a preliminary assessment of the abuse liability of these cathinone analogs. We agree that long-term studies are essential to understand chronic impacts and potential neurotoxicity. We plan to pursue longitudinal studies in mammalian models in the future.

Comment 5: Pharmacological Differences: The study mentions the differences in pharmacological actions, such as DAT/SERT selectivity and SERT substrate releasing properties. However, it does not delve deeply into how these differences might affect other aspects of drug abuse liability, such as tolerance, sensitization, or cross-sensitization with other drugs.

Response 5: As suggested, we added the effects of tolerance of SERT on abuse potential on page 8 as follows.

Moreover, the abuse potential of mixed-action synthetic cathinones has been shown to increase with repeated exposure, possibly due to the sustained DAT-mediated effect while tolerance develops to SERT-mediated effects that initially limit abuse potential of these compounds [27].

Round 2

Reviewer 1 Report

Comments and Suggestions for Authors

The Authors provided require corrections, thus improved the paper. I suggest its publication